# Analyzing the Factors of Vacant Home Occurrence for Urban Sustainability: A Case Study of Medium-Sized Cities Focusing on Asan City, Chungcheongnam-do

**Jeong-hyeon Choi \*, Seung-Seok Han and Myung-je Woo \***

Department of Spatial & Environmental Planning, ChungNam Institute in Republic of Korea, Gongju 32589, Republic of Korea; ssh@cri.re.kr
\* Correspondence: starcjh82@gmail.com (J.-h.C.); mwoo@uos.ac.kr (M.-j.W.); Tel.: +82-10-8837-6856 (J.-h.C.); +82-2-6490-2803 (M.-j.W.)

**Abstract:** This study aims to enhance urban sustainability by analyzing the spatial distribution and underlying causes of vacant homes in Asan, Chungcheongnam-do. Various statistical methods were employed to analyze date concerning the number of vacant stores, population changes, land use complexity, and the physical characteristics of land and buildings; these factors were found to influence the prevalence of vacant homes across Asan. Additionally, it was found that distinct factors differentially affect specific regions, such as old downtown areas versus rural villages. This indicates that reducing vacant homes in these areas requires distinct policies tailored to the unique circumstances of each region. For instance, in old town areas, small lot ratios and land use complexity are significant, while in rural villages, the average number of floors and land use complexity play a major role. This study highlights the diverse factors influencing the prevalence of vacant homes and suggests that to effectively address this issue, policies should be developed that are tailored to the unique characteristics of each area, categorized at both the city and local levels.

**Keywords:** vacant homes; urban sustainability; Poisson regression; negative binomial regression; spatial analysis

## 1. Introduction

Along with economic growth in South Korea, the population began to become concentrated in urban areas; the urbanization rate reached 91.8% in 2019. However, currently, not only South Korea but also many countries that are considered to be economically advanced are facing serious social issues such as low birth rates, aging populations, and low economic growth. The impact of these issues became more pronounced after the COVID-19 pandemic. The Ministry of Public Administration and Security emphasized that 2020 marked the beginning of a population decline, a surge in one- and two-person households, and the lowest birth rate ever, necessitating changes across all social and economic sectors. In reality, the majority of domestic cities are experiencing population decline (as of 2020, only 5 out of 17 provinces and cities saw population growth), and both urban and non-urban areas are witnessing an increase in abandoned and unused land and buildings. In 2019, 322 construction projects were abandoned, and there were 1,517,815 vacant homes (National Statistical Portal) and 3829 closed schools (Ministry of Education). Historical downtown areas are seeing an increase in abandoned property as the population, industry, and public institutions move to new developments on the outskirts of cities. Additionally, the number of vacant stores has surged due to the impact of COVID-19. Given these structural social changes and changes in urban resource demands, urban planning needs to move away from the traditional focus on quantitative expansion and vertical/horizontal extension. Instead, there is a need for the short-term efficient management and recycling of existing resources and a long-term shift toward compact and mixed-use development management.

An increase in vacant homes not only leads to social and economic issues, such as increased crime and decreased property values, but also to urban decline, negatively affecting the surrounding residential environment [1–5].

In recent years, medium-sized cities like Asan in Chungcheongnam-do have experienced an alarming increase in the numbers of vacant homes, a trend that exacerbates urban decline and poses significant challenges to urban sustainability. This study addresses the urgent need to understand the spatial distribution and causal factors of these vacancies, especially in the post-COVID-19 era, where urban dynamics have markedly shifted. By integrating advanced spatial analysis with socio-economic data, our research fills a significant gap in the urban studies literature, providing nuanced insights that can guide effective urban policy. The purpose of this study is to spatially analyze the characteristics of the increasing number of vacant homes at the municipal and neighborhood levels, stemming from socio-economic changes such as low birth rates, aging populations, development polarization, and population outflows from regional cities. This study aims to identify factors contributing to the prevalence of vacant homes, employing a comprehensive urban perspective, and to analyze how these factors vary according to the local neighborhood conditions. By doing so, we seek to offer insights into the issue of vacant homes in small and medium-sized cities, considering the diverse and changing socio-economic landscape. The present findings are expected to provide a foundation for sustainable urban development strategies that are acutely needed for medium-sized cities facing similar challenges across South Korea and beyond. This research not only enriches the academic discourse on vacant homes but also serves as a crucial tool for policymakers aiming to revitalize these vital urban areas.

For the purpose of this research, which seeks to analyze the factors leading to the occurrence of vacant homes, the study was conducted with three areas of focus. First, the relationship between urban decline and the occurrence of vacant homes was considered. Prior studies have identified vacant homes as both a cause and an outcome of urban decline, indicating that an increase in the number of vacant homes is closely related to urban decline [6,7]. It has been argued that one of the typical phenomena resulting from urban decline is the emergence of vacant homes [8]. Moreover, the pattern of vacant homes can serve as an important indicator of shrinking cities [9].

Second, the study considered the differences in urban structure and development characteristics between large cities and small cities. A distinctive feature of urban development in Korea is the universal issue of downtown decline due to suburbanization. In most cities across the country, residential development has been carried out on the outskirts of the city [10]. However, the pattern of downtown decline differs between large cities and small cities. In large cities, industrial development along with increased labor demand and productivity improvements have led to the creation of business agglomerations, increasing the availability of goods and services. This, coupled with the convenience of various facilities, has attracted residents and businesses from surrounding cities. On the other hand, small cities have experienced a continuous decrease in population and industry due to a reduction in agricultural demand caused by advancements in agricultural technology, a small number of industrial facilities, and poor living and business environments [11].

Third, the study identified that the factors and the impacts regarding vacant home occurrences differ between large cities and small cities. In large cities, homes often become vacant due to speculative purchases made with the intention of profiting from the designation or cancellation of redevelopment areas. Even if there is an outflow of population and industry from the original downtown areas due to new town developments, the impact is lessened because of the influx from surrounding cities. Additionally, large cities have a high diversity in their industrial structure and a significant proportion of high-value-added, advanced knowledge industries, which lowers the likelihood of continuous urban decline [12]. However, in small cities, the smaller scale of the housing market can lead to a chain reaction of vacancies due to oversupply, and there is a high risk of this situation worsening due to a decrease in population and demand. Fundamentally, the lack of demand and feasibility

makes it difficult to push forward with redevelopment projects, leading to an increase in the prevalence of vacant homes.

## 2. Selection of Research Target

The selection of the research site was the first step in this study. We sought to find a representative area among South Korea's medium-sized cities that could suggest policy directions for solving the vacant home problem. The selection criteria were as follows: First, similarity in development characteristics and urban form. Priority was given to medium-sized cities that share similar development characteristics and urban forms with South Korea. This was to increase the generalizability of the research results. Second, representativeness of medium-sized cities. Through the analysis of various indicators, the aim was to select a city that could represent South Korea's medium-sized cities. This was to specify the research scope and facilitate an in-depth analysis of the vacant home problem. Considering these criteria and the presence of a vacant home information system, Chungcheongnam-do was initially selected, excluding Gyeonggi-do and Jeju-do from the metropolitan governments. Among the 15 cities and counties in Chungcheongnam-do, both Asan and Seocheon showed population distribution Gini coefficients similar to the national average. Furthermore, upon reviewing local statistics centered on indicators identified in prior research as affecting the occurrence of vacant homes, Asan city was ultimately found to match the characteristics of a medium-sized city in numerous indicators. The period of study targeted the year 2020, i.e., prior to the impacts of the COVID-19 pandemic.

## 3. Literature Review and Theoretical Framework

### 3.1. Background and Factors Causing Vacant Homes

Cities are the foundation for populations to live, containers for their activities, and spaces where the future of the population can be anticipated. The issues arising from population changes are diverse, especially those occurring in physical spaces, which have a more profound perceived impact. In Korea, population overconcentration in metropolitan areas and accelerated outflow from provincial areas, along with imbalanced national land development, have led to the problems of population and industry exodus, centered around old downtown areas in regional cities. As a result, vacant properties, such as closed schools, abandoned industrial facilities, defunct stations, vacant homes, and empty stores have emerged.

As shown in Table 1, vacant and abandoned properties are defined in various ways across studies, with terms like "vacant land" and "empty home" being most commonly used. The American Planning Association (APA) defines "vacant land" as land or buildings that are not actively used for any purpose, or plots of land without any improvements [13]. The term "abandoned property" encompasses a broader concept, including physical neglect and the relinquishment of tax and economic value. According to the Legal Information Institute, an abandoned property is defined as private property that the owner has intentionally forsaken all control over rights to. Other terms include "lost space," representing structural and functional loss, "blighted property," referring to land that has become desolate and abandoned, "derelict land" and "brownfield" for polluted and neglected lands, "TOADS" for contaminated lands impacting surrounding areas, and "void" for abandoned spaces in architecture. This study focuses on vacant homes which are privately owned and occur frequently in both urban and rural areas, presenting challenges to urban development.

As shown in Figure 1, vacant homes are the result of various factors such as population, industry, economy, culture, neighborhood environment, and their physical condition acting on a space in a complex. Previous studies have classified the factors leading to the occurrence of vacant homes into four categories. First, there are urban growth and morphological factors. This perspective views vacant properties as a result of urban growth processes and changes in urban form and spatial structures. In the early 1900s, sociologists at the University of Chicago, centered around Ernest W. Burgess and known as the Chicago School, established an ecological perspective on urban activity changes, adaptation, and

competition, which they analyzed in terms of urban spaces. Similarly, M. R. Conzen differentiated streets, plots, and buildings as factors constituting urban form, and viewed the city as changing its composition and organization through the correlation, interaction, and recombination of these elements [14].

**Table 1.** Classification of abandoned property types according to previous research.

| | International Research | Korea Research | |
|---|---|---|---|
| **Division** | **Definition** | **Public Owned** | **Private Ownership** |
| Vacant land, empty home | physically neglected and unmanaged properties with little to no use | former facilities, abandoned stations, closed schools | vacant homes and stores, undeveloped plots |
| Abandoned property | absence of occupancy, structural and functional damage, and the owner's abandonment of economic responsibility | - | - |
| Lost space | spaces that are not effectively utilized and contribute nothing to their surroundings | traffic islands, spaces under bridges or roads | - |
| Blighted property | similar to vacant and abandoned properties but emphasizes physical damage and resultant harm to the surrounding area | - | - |
| Derelict land, Brown fiend | land that has been polluted and abandoned due to its previous industrial or commercial use | military base relocation sites | inoperative industrial facilities, abandoned mines |
| TOADS | temporarily obsolete and deserted lands, similar to derelict lands but with an emphasis on the impact on surrounding lands | - | - |
| Void | spaces that are out of sync with their surroundings, not recognized or used | - | old industrial facilities and Asan Rose Village |
| Under utilized property | assets that are not used to their full potential or only used intermittently or irregularly | Cheonan Asan Station plaza | older industrial parks or shopping districts |

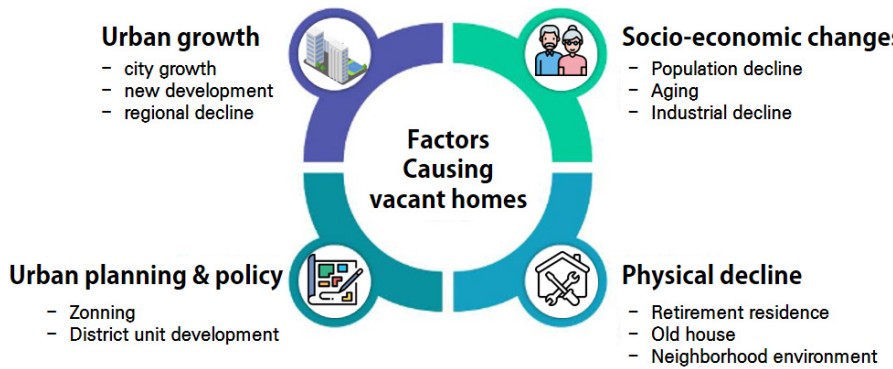

**Figure 1.** Factors Causing vacant homes.

Second, socio-economic factors. The occurrence of vacant homes is influenced by changes in social and economic conditions such as population decline, aging, decline in local industries, and the new opening or relocation of infrastructure (roads, railways, stations, etc.). Among socio-economic factors, the most significant impact on the occurrence of vacant homes is due to population decline and aging. Since the mid-1970s, the urban population has surpassed the rural population, and starting from the 2000s, the issue of vacant homes in rural areas began to emerge, leading to the initiation of measures to address this problem. The decline in local industries is also an important factor. It has been shown that abandoned and vacant homes are scattered in areas where the fishing industry has declined, such as Mukho Port in Donghae City and Jeongrajin Port in Samcheok City, and

in areas where coal mines have closed, such as Cheoram in Taebaek City and Dogye-eup in Samcheok City [15].

Third, urban planning and policy factors. Vacant homes can occur due to urban planning and related laws and regulations, as well as policy implementation by central and local governments. Korea limits the use, coverage ratio, and floor area ratio of buildable buildings through the zoning system and designates land use districts and other measures to complement these restrictions. Additionally, for the purpose of preserving certain areas for specific goals, use zones are designated and managed. Buildings located in specific use districts or zones may face restrictions on construction activities such as additions or reconstruction, leading to difficulties in maintenance and eventually to abandonment. The district unit plan is a scheme that can partially strengthen or relax restrictions on land use to rationalize land use and enhance its function. However, if a district unit plan is decided upon when there are existing buildings, vacant homes may occur if the homeowners or purchasers find it difficult to develop according to the district unit plan.

Fourth, neighborhood environment and physical factors. These can be considered the most direct factors leading to the occurrence of vacant homes. In Korea, the issue of vacant homes began to emerge in rural areas. Studies have shown that vacant homes and abandoned properties occur in aging residential areas in the provinces [16] that are prevalent in aging residential areas constructed on poor infrastructural hillside locations, where a relatively high proportion of the elderly population is evident, and buildings are often left abandoned after the death of the elderly residents [15]. A study on the factors leading to property neglect in New York City found that the quality of property maintenance, neighborhood vacancy rates, and the condition of neighboring houses are significant factors contributing to the occurrence of vacant homes [17]. It was determined that property neglect occurs less in areas with a lower rate of housing decay [18].

### 3.2. The Negative Impact of Vacant Homes on the Region

Numerous prior studies have indicated that vacant homes impact the surrounding neighborhood areas [19,20]. The most renowned theory in this context is the "Broken Window Theory". This theory, proposed by American criminologists James Q. Wilson and George L. Kelling in 1982, posits that leaving a broken window unattended can lead to a spread of crime centered around that spot. The occurrence of vacant homes initially leads to physical issues such as fires, diseases, crime, safety hazards, and the deterioration of the living environment. Subsequently, this can lead to urban blight, including the degradation of city aesthetics and population exodus (Figure 2). However, one of the core aspects of the "Broken Window Theory" is that the problems resulting from physical damage can spread across various sectors. Therefore, the theory emphasizes that swift action is necessary to address the issue of vacant homes in order to prevent the waste of administrative efforts and public funds.

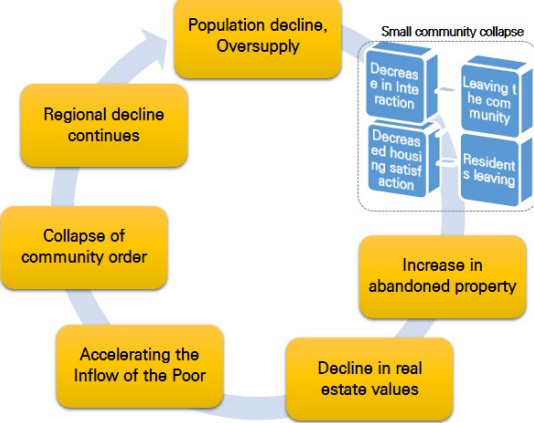

**Figure 2.** Negative effects of Vacant homes.

*3.3. Theoretical Framework*

The history of research related to vacant homes is quite extensive, and during this period, various types of studies have been conducted. Initial research focused on the distribution of vacant homes and the urban problems that arise from them [21], and gradually expanded to include studies on causes, management strategies, and utilization methods. Research on the distribution of vacant homes has been conducted in urban and rural areas, analyzing spatial distribution patterns and using logistic regression analysis. Studies on the factors of vacancy have utilized various explanatory variables depending on the target city, such as large cities, medium-sized cities, and rural areas, and applied models like the LASSO model, zero-inflated negative binomial regression model, multilevel model, and geographically weighted regression model, according to local conditions [22]. Research has also been conducted to prevent the occurrence of vacant homes and to manage them in a planned manner [19]. Lastly, studies have been carried out considering the characteristics of vacant homes concentrated in declining areas, including urban regeneration using vacant homes and residential support schemes for artists [23,24]. The primary objective of this study is to analyze the factors leading to the occurrence of vacant homes. However, unlike previous research, this study focuses on demonstrating the differences between the causes at the city level and the causes specific to certain areas.

To analyze the causes of vacant homes, it is necessary to review theoretical backgrounds on the growth, development, and decline of cities, national economic growth policies, and characteristics such as urban growth dynamics. The theoretical establishment of a city's growth process, based on changes in the city's size, population, and functions, is known as the urban development stage theory. The theory of urban development stages focuses on the creation of economic opportunities, technological advancements, and urban expansion, and has recently expanded to include the population growth across different urban areas. Stages of urban development model divided the process of urban development into four stages and eight periods [25]: 1st stage urbanization, 2nd stage suburbanization, 3rd stage desurbanization, and 4th stage reurbanization. According to the theory, as a city goes through urbanization to reach the suburbanization stage, the rate of population movement from urban to suburban areas exceeds the natural increase rate, leading to a stage of absolute dispersion where the urban population decreases and the suburban population increases. Then, reaching the desurbanization stage, the city undergoes a stage of population dispersion across the entire urban area. From an overall urban perspective, stages 1 and 2 are seen as periods of population increase, while stages 3 and 4 are viewed as periods of decrease. Asan city, for instance, is in the 2nd stage, 4th period, where the overall city population is increasing, but the urban center population is decreasing, and the population in the outskirts and suburban areas is increasing.

Prior studies have shown that vacant homes arise as a result of urban decline and that the increase and entrenchment of vacant homes can serve as a catalyst for further decline. According to a survey on vacant homes in Daegu Metropolitan City, 18.9% of respondents indicated that urban decline, such as the aging of urban infrastructure, was the cause of vacant homes [26]. Furthermore, studies have demonstrated that vacant home clustering in declining areas can accelerate regional decline. Thus, it is necessary to include factors such as the aging of urban infrastructure and indicators of urban decline when analyzing the causes of vacant homes.

Cities in South Korea possess diverse characteristics based on their size, function, and location. More than 50% of the total population resides in the metropolitan area of Seoul, the country's primary city. National infrastructure and industries are concentrated along the axis connecting Seoul and Busan. There are metropolitan cities and special cities with populations of over one million, metropolitan areas consisting only of urban regions, mixed urban–rural cities in the provinces, and agriculturally functional small towns and rural districts, reflecting a wide variety of conditions. Due to the differing conditions across cities, research on the causes of vacant homes has been conducted at various levels, including nationwide, in large cities, and in regional medium-sized cities. This research trend has

also been conducted in a similar manner in Japan [27]. Furthermore, within the same city, the causes of vacant homes can vary depending on the local conditions of different areas. The occurrence of vacant homes is influenced by detailed data on buildings and land, as well as the distribution of major facilities located nearby [28]. Many prior studies have focused on the spatial autocorrelation of vacant home occurrences, analyzing the causes based on spatial data [29,30]. This study focuses on Asan City as a case study to analyze the causes of vacant homes, considering the spatial distribution of vacant homes. It analyzes the causes of vacant home occurrences from both a city-wide perspective and a localized perspective.

## 4. Materials and Methods

### 4.1. Variable Settings

The dependent variable, vacant homes, refers to homes that have been empty for over a year according to relevant laws. A survey on the state of vacant homes is conducted at the city level by the Korea Real Estate Board, an organization supporting the management of vacant homes. The survey list is extracted based on the usage of energy and water. The process involves selecting investigators, interviewing owners, and conducting on-site surveys to ultimately determine the number of vacant homes. The surveyed number of vacant homes is used in the planning of vacant home management and urban regeneration projects. The detailed procedure is shown in Figure 3. According to the 2020 survey conducted by Asan City and the Korea Real Estate Board, there were 266 vacant homes in Asan City. Due to the potential distortions in statistical surveys caused by the COVID-19 pandemic starting from 2021, data from 2020 will be used.

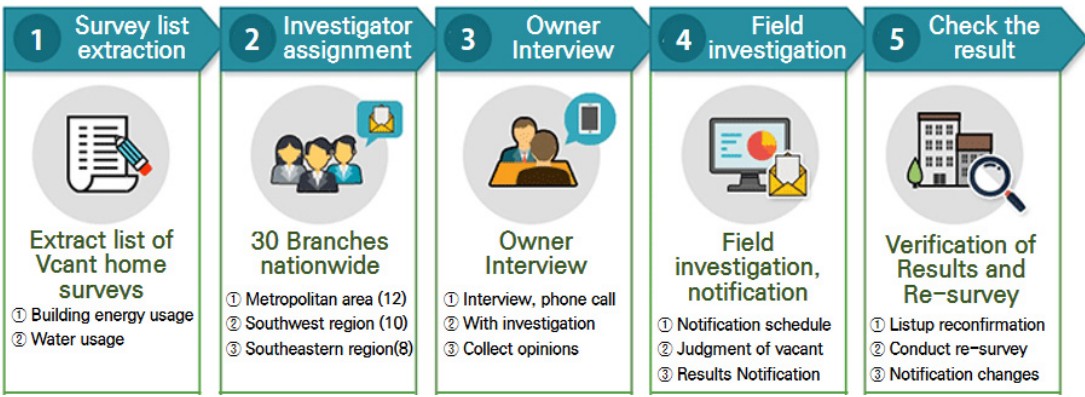

**Figure 3.** Vacant Home Survey Procedures (Korea Real Estate Board).

A review of previous research showed that many variables have been considered in determining the factors that influence the occurrence of vacant homes, depending on the purpose of the research. As shown in Table 2, These factors can be divided into internal factors, which consider the characteristics of the building or land itself, and external factors, such as the physical environment and social conditions surrounding the property. Small and irregular plots, as well as poor accessibility, are commonly seen in old residential areas or old downtowns, and there is generally a tendency for people to avoid these poor living conditions as economic development and living conditions improve. Therefore, land characteristics that have a direct correlation with the occurrence of vacant homes were selected as explanatory variables. With improved access to data through public data platforms, the use of building registry information has increased. In areas near downtowns where vacant homes are commonly found, the average building floor area is small, structures other than reinforced concrete are common, and the ratio of single-story buildings is high. Considering these characteristics, average floor area, specific building structure, single-family home ratio, and average number of floors were utilized. Among external factors, the most explanatory variables were selected from the physical

environment surrounding the vacant homes. Housing ratio, distance to main roads, road accessibility, new development projects, and the progress of redevelopment projects are representative indicators of the physical environment of an area. Elevation and slope are indicators showing the development conditions of the land, and the ratio of new to old homes is an indicator that can analyze the investment situation in the area. Distance to the regional center, main road accessibility, and highway interchange accessibility can be used to analyze the level of transportation services in the area, and the distance to new development project sites is an indicator to reflect the impact of nearby development projects. Considering the industrial economy sector, Asan City is an industrial city based on manufacturing; manufacturing and construction play a crucial role in the city's economic development. Thus, the number of workers in manufacturing and construction and their rate of change were selected as variables. In the population and social sector, the ratio of women of childbearing age and the elderly population ratio have been shown to be related to the occurrence of vacant homes in numerous studies and were used in our study to predict the sustainability and decline of the area. Floating population is one of the most frequently used parameters to assess the vitality of an area. To understand changes in housing demand, population density, total population change rate, and the rate of change in the elderly population were selected. Lastly, the value of a home is determined by its residential utility [31], which is influenced by factors determining residential location [32]. The central government publishes national land surveys that diagnose changes in living conditions in grids, among which access to elementary schools, cultural facilities, and hospitals were selected as explanatory variables in the living Social Overhead Capital (SOC) area.

**Table 2.** Variable composition and explanation.

| Category | | | Variable Name | Description |
|---|---|---|---|---|
| Dependent Variable | | | Vacant Homes (units) | Survey on vacant homes by Asan City |
| Explanatory Variables | Internal Factors | Land Conditions | Undersized Plots (%) | Percentage of plots less than 60 m$^3$ |
| | | | Irregularly Shaped Plots (%) | Percentage of irregularly shaped plots |
| | | | Poor Access Plots (%) | Percentage of plots inaccessible by vehicle |
| | | | Public Land Value (KRW) | Individual public land value as of January 2020 |
| | | | Land Use Diversity | Ratio of different types of buildings within a grid |
| | | Building Conditions | Average Floor Area (m$^3$) | Average floor area of buildings |
| | | | Specific Building Structures (%) | Percentage of block, wood, stone structures |
| | | | Detached Houses (%) | Percentage of detached houses |
| | | | Average Number of Floors | Average number of floors in buildings |
| | External Factors | Physical Environment | Main Road Accessibility (m) | Accessibility to main roads wider than 15 m |
| | | | Elevation (m) | Average elevation |
| | | | Slope (°) | Average slope |
| | | | Highway IC Accessibility | Distance to the nearest highway interchange |
| | | | New Houses (%) | Percentage of houses built within the last 5 years |
| | | | Old Houses (%) | Percentage of houses older than 20 years |
| | | | New Buildings (%) | Percentage of buildings built within the last 5 years |
| | | | Development Permit Activity | Number of development permits issued |
| | | | Distance to Regional Centers (m) | Distance to primary nucleus and five regional hubs |
| | | | Distance to New Development (m) | Distance to residential development project sites |
| | | | Building Density | Number of buildings per hectare |

**Table 2.** *Cont.*

| Category | | | Variable Name | Description |
|---|---|---|---|---|
| Explanatory Variables | External Factors | Industrial Economy | Number of Manufacturing Workers | Number of workers in manufacturing in 2020 |
| | | | Manufacturing Workers Change Rate (%) | Change rate between 2010 and 2020 |
| | | | Number of Construction Workers | Number of workers in construction in 2020 |
| | | | Construction Workers Change Rate (%) | Change rate between 2010 and 2020 |
| | | | Vacant Stores (units) | Public data utilization estimate |
| | | Population and Society | Fertile Women (%) | Percentage of women aged 20–40 |
| | | | Elderly Population (%) | Percentage of population aged 65 and above |
| | | | Total Floating Population | Floating population within a 50 m grid |
| | | | Elderly Floating Population | Floating population of elderly within a 50 m grid |
| | | | Floating Population Change Rate (%) | Change rate between October, 2019 and 2020 |
| | | | Population Density (people/ha) | Number of people per hectare |
| | | | Total Population Change Rate (%) | Change rate over the last five years |
| | | | Elderly Population Change Rate (%) | Change rate of elderly population last five years |
| | | | Aging Index | Population aged 65/population under 15 years |
| | | Living SOC | Elementary School Accessibility (m) | Distance to the nearest elementary school |
| | | | Cultural Facility Accessibility (m) | Distance to cultural facilities |
| | | | Hospital Accessibility (m) | Distance to hospitals |

*4.2. Analysis Methods and Procedures*

The analysis conducted in this study involved investigating the factors that contribute to the occurrence of vacant homes throughout Asan City and examining the reasons for vacancies at a localized level based on the characteristics of different areas. To begin with, we conducted a spatial analysis of the current status of vacant homes in Asan City to examine patterns of spatial concentration and dispersion. The spatial analysis unit proceeded with a 500 m × 500 m grid, reflecting an appropriate radius of 500 m for a density analysis at the micro-level within urban areas [33,34]. Three types of analysis were conducted for spatial distribution. The first was Quadrat analysis, which analyzes the density of points within a grid that has been divided into regular intervals [35]. The second was Kernel Density Estimation. The Kernel Function is non-negative and symmetric around the origin and has an integral value of one. The density value calculated from the point to the set distance is reassigned to the grid to identify hot spots. Lastly, according to the First Law of Geography [36], everything is related, but things that are closer are more related than those that are further away. If the subject, which is assumed to have a cause and effect, shows a systematic pattern in space, it can be considered to have spatial autocorrelation; therefore, we utilized the commonly used Moran's I statistic for measurements.

Following the identification of regions with high concentrations of vacant homes through a spatial distribution analysis, the areas were standardized for typification. A cluster analysis was conducted utilizing the attribute values of the explanatory variables to classify these standardized areas. The types derived from the cluster analysis results were named considering the locational conditions and attribute values. Finally, the localized impact of vacant home occurrences was analyzed based on the classified types.

The explanatory variables presented in Section 4.1 were selected as all possible variables that were expected to influence the occurrence of vacant homes in Asan City. To identify the factors that actually affect vacant homes, we conducted a correlation analysis. Pearson correlation coefficient and Spearman correlation coefficient are commonly used in correlation analysis. Spearman's rank correlation coefficient measures the correlation based on ranked values of each variable. Therefore, in this analysis, the Pearson correlation coefficient was used to quantify linear correlations. While correlation analysis explains the relationships between variables, it does not explain cause and effect. Typically, in social

sciences, regression models are used to explain causal relationships between variables. Due to the large number of explanatory variables anticipated to influence the occurrence of vacant homes, multiple regression models were utilized. Linear regression models require assumptions of linearity, independence, homoscedasticity, and normality. However, the spatial clustering identified in the preliminary spatial analysis of vacant homes indicated that the assumptions of homoscedasticity and normality were not satisfied. Therefore, spatial regression models, which account for spatial similarities and differences, were employed for more accurate model specification and analysis. Prominent among spatial regression models are the Spatial Lag Model (SLM) and the Spatial Error Model (SEM). Generally, if spatial dependence is present in the dependent variable, it is appropriate to use the SLM, which incorporates the influence of neighboring observations by adding spatially lagged independent variables into the model. Specifically, it uses the average of neighboring area variables to construct a spatial weights matrix, which is then included in the model as the spatial lag variable (Wy). In this study, adjacency is determined using the Queen criterion to construct the spatial weights matrix. If spatial autocorrelation is found in the errors of a multiple regression model result, it is suitable to use the SEM, which controls for spatial dependence among errors by creating a covariance among them to consider spatial interactions within the regression model. The SEM applies a spatial weights matrix to the errors. The suitability of these two models is assessed during the empirical analysis phase using the Lagrange Multiplier Test (LM), with LM-Lag and LM-Error statistics helping to select the appropriate spatial regression model. The dependent variable, the number of vacant homes, represents the frequency of occurrence and cannot take negative values. It is important to construct a model with a function capable of appropriately reflecting the distribution of the count data to estimate the impact of the explanatory variables. Typically, for events where the probability of occurrence is very low, resulting in many zeros, the Poisson model is used, which assumes equality between mean and variance. However, the preliminary spatial analysis in Asan City revealed a high number of grids with no vacant homes, indicating that the assumption of equal mean and variance was not met. If overdispersion occurs where the variance of count data exceeds the mean, the negative binomial regression model may be more appropriate. Therefore, both the Poisson and negative binomial regression models were additionally utilized. This concludes the analysis of factors causing vacant homes in Asan City from a global perspective. Lastly, Geographically Weighted Regression was conducted to estimate the factors for vacant home occurrences for each typified region.

## 5. Analysis Results

### 5.1. Spatial Statistical Analysis and Type Classification

Based on the vacant home data from Asan city, a quadrat analysis was conducted using ArcGIS. As shown in Figure 4, The entire area of Asan city was divided into 500 m × 500 m grids, and among these, 1844 grids containing buildings were analyzed. Generally, if the variance-to-mean ratio equaled 1, it was interpreted as regular; if less than 1, as random; and if greater than 1, as clustered [37]. The analysis results of vacant homes showed a variance-to-mean ratio of 5.05, indicating that vacant homes in Asan city are clustered. The Global Moran's I statistic for vacant homes in Asan city is 0.075, indicating a low level of spatial autocorrelation overall. The results of the Local Moran's I analysis for deriving clustered areas of vacant homes revealed the distribution of HH type clusters in certain areas, including the old downtown centers around Onyang1-dong and Onyang2-dong, as well as in Baebang-eup and Tangjeong-myeon.

To standardize areas with a high concentration of vacant homes and derive clustered areas, as well as to categorize regions with similar characteristics, a cluster analysis was performed. K-means clustering analysis was conducted to minimize the variance within each group. K-means clustering is one of the machine learning algorithms that groups data points into multiple clusters. It classifies data points based on the centers of each cluster to create groups with high homogeneity. The dendrogram analysis indicates that

dividing into two clusters is appropriate. Cluster 1 was characterized by a high ratio of poor road connectivity, low individual land prices, low building density, and low pedestrian traffic. It also has low accessibility to main roads and a high proportion of block, log, and stone structures, defining it as a "Rural Village type". Cluster 2 was defined by a large average floor area, high average number of floors, and high individual land prices. It has a high ratio of detached houses, high building and population densities, and a large elderly population, defining it as an "Old Town type". The final type is as shown in Figure 5.

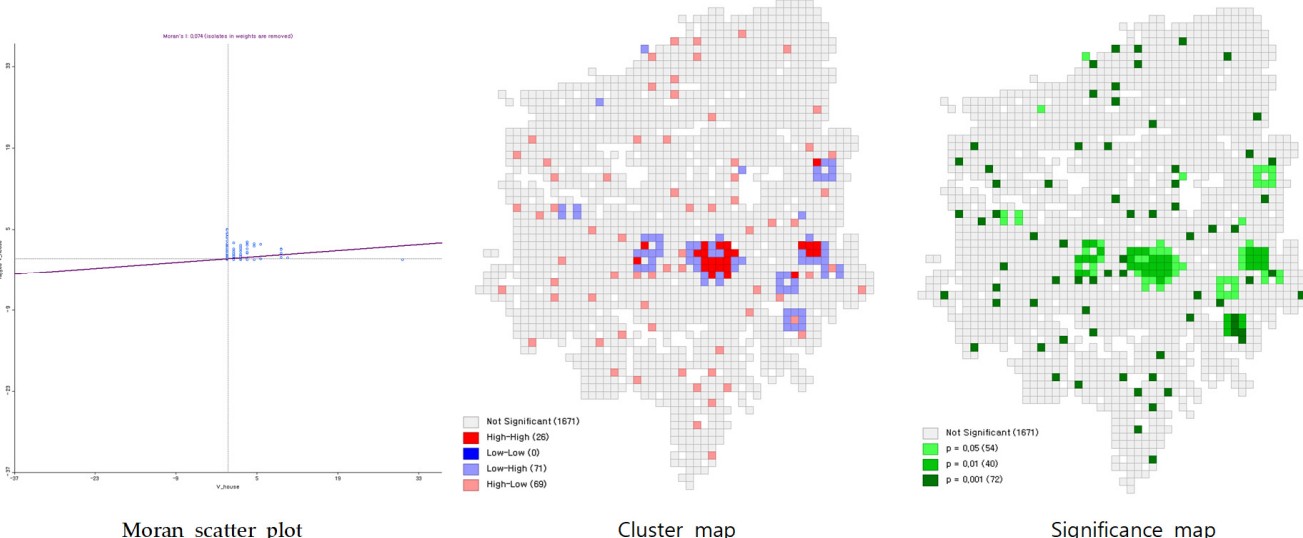

| Moran scatter plot | Cluster map | Significance map |

**Figure 4.** Results of spatial autocorrelation analysis of Vacant homes in Asan city.

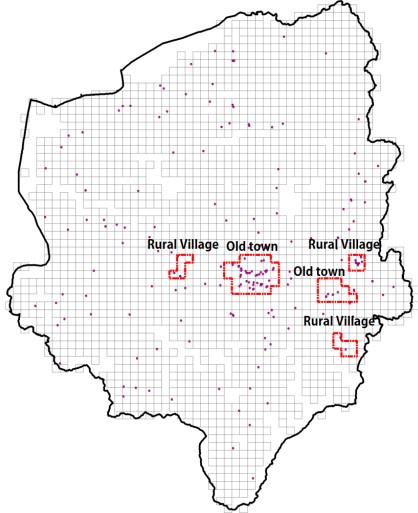

**Figure 5.** Types of vacant home clusters.

*5.2. Factors Causing Vacant Homes in Asan City from a Global Perspective*

To analyze the factors affecting the occurrence of vacant homes in Asan city from a global perspective, the correlation between 36 explanatory variables and the number of vacant homes was examined. When there are too many explanatory variables, issues such as model overfitting and multicollinearity can arise, necessitating the elimination of explanatory variables through correlation analysis. Pearson correlation analysis, the most commonly used method for analyzing the correlation between interval and ratio scale variables, was conducted. The final explanatory variables selected after excluding variables with significant correlation at the 0.01 level and suspected of multicollinearity were 'vacant stores', 'undersized plot ratio', 'average number of floors', 'total population change rate',

'distance to new development projects', 'number of construction workers', and 'land use mix', totaling seven.

To select the optimal analysis model for the factors influencing the occurrence of vacant homes, the Lagrange Multiplier Test for the general regression model was conducted. The Robust LM (lag) statistic was 12.468, and the *p*-value was 0.0032, rejecting the null hypothesis of no spatial lag autocorrelation at the 0.01 significance level. This indicated the presence of strong spatial lag autocorrelation. Therefore, it was verified that the spatial lag model was more suitable for analyzing the factors influencing vacant home occurrences in Asan City.

The results of the general regression model, spatial lag model, and spatial error model are as follows (Table 3). The Log Likelihood statistic was highest for the SLM model, indicating the best fit to the data among the models tested. However, considering the balance between model fit and complexity, as indicated by the AIC and SC values, the SLM model did not necessarily perform the best, suggesting that the increase in model complexity did not proportionally increase the fit. Given the diverse conditions within Asan city, selecting a model based solely on simple statistical comparisons may not always be the best approach. The coefficient of determination was highest for the SLM, meaning it explained the variability in the observed data slightly better than the other models and captured the data structure more accurately by including spatial autocorrelation. However, the relatively low coefficient of determination values for all models suggested that the explanatory power of the models was limited, indicating the need for additional variables or changes to the model.

**Table 3.** Analysis results of OLS, SLM, SEM.

| Category | OLS | SLM | SEM |
|---|---|---|---|
| Constant | −0.01911 | −0.02913 | −0.01948 |
| Vacant Stores | 0.191 *** | 0.035 *** | 0.036 *** |
| Undersized Plot Ratio | 0.026 | 0.333 | 0.341 |
| Average Number of Floors | 0.071 *** | 0.047 *** | 0.048 *** |
| Total Population Change Rate | 0.060 *** | 0.004 *** | 0.004 *** |
| Distance to New Development | −0.047 ** | −0.00001 * | −0.00001 * |
| Number of Construction Workers | −0.023 | −0.001 | −0.001 |
| Land Use Mix | 0.091 *** | 0.029 *** | 0.030 *** |
| Spatial Lag Coefficient (Rho) | - | 0.0632 | - |
| Spatial Error Coefficient (Lambda) | - | - | 0.04111 |
| R-squared | 0.0800 | 0.0854 | 0.0843 |
| Log Likelihood | −2247.38 | −2246.16 | −2246.90 |
| AIC | 4500.77 | 4510.32 | 4509.8 |
| SC | 4554.93 | 4559.99 | 4553.96 |
| Jarque-Bera | 22,857,397 *** | | |
| Breusch-Pagan | 1574.07 *** | 1569.63 *** | 853.576 *** |

*: *p*-value < 0.1, **: *p*-value < 0.05, ***: *p*-value < 0.01.

The research analysis unit of a 500 m grid encompassed a total of 1844 grids across Asan city, but since 1690 of these grids had a dependent variable of vacant homes equal to zero, the distribution was skewed toward zero rather than following a normal distribution. Since the probability of occurrence was very low, resulting in many zero outcomes, and because the number of vacant homes was a form of countable data, further analyses were conducted using Poisson regression models and negative binomial regression models. As shown in Table 4, The comparison of Log Likelihood, AIC, and BIC statistics showed that the negative binomial regression model had a greater impact on some variables compared to the Poisson regression model and also showed superior performance metrics. It had a higher explanatory power compared to the previously analyzed OLS, SLM, and SEM models. The analysis results from the negative binomial regression model revealed that the

factors influencing the occurrence of vacant homes in Asan city from a global perspective are 'vacant stores', 'undersized plot ratio', 'average number of floors', 'total population change rate', and 'distance to new development'. It was found that an increase of one unit in vacant stores led to an increase of 0.03 homes being vacant. This indicated a strong positive correlation between vacant homes and vacant stores, both of which are idle real estate resulting from urban decline. It was found that for every additional floor, the number of vacant homes increased by 0.17. This is contrary to previous research, which typically shows that areas with high-density development have a lower occurrence of vacancies due to higher demand [38]. This phenomenon was interpreted as being due to the concentration of vacant houses in areas with old apartments, located in both the urban core and suburban regions.

**Table 4.** Analysis results of PRM, NBRM.

| Category | Poisson Regression | | | Negative Binomial Regression | | |
|---|---|---|---|---|---|---|
| | β | Exp (β) | *p* | β | Exp (β) | *p* |
| Constant | −2.082 | 0.125 | 0.000 | −2.358 | 0.095 | 0.000 |
| Vacant Stores | 0.016 | 1.016 | 0.014 | 0.029 | 1.029 | 0.011 |
| Undersized Plot Ratio | 4.961 | 142.807 | 0.000 | 5.393 | 219.839 | 0.000 |
| Average Number of Floors | 0.091 | 1.095 | 0.003 | 0.176 | 1.193 | 0.002 |
| Total Pop Change Rate | 0.006 | 1.006 | 0.001 | 0.005 | 1.005 | 0.045 |
| Distance to New Development | −0.00009 | 1.000 | 0.000 | −0.00007 | 1.000 | 0.003 |
| Num. of Construction Workers | 0.005 | 1.005 | 0.157 | 0.004 | 1.004 | 0.399 |
| Land Use Mix | 0.003 | 1.003 | 0.924 | −0.008 | 0.992 | 0.817 |
| Deviation | | 0.802 | | | 0.565 | |
| Pearson Chi-Square | | 1.924 | | | 1.570 | |
| Log Likelihood | | −566.505 | | | −503.687 | |
| AIC | | 1149.009 | | | 1023.375 | |
| BIC | | 1188.752 | | | 1071.118 | |

### 5.3. Factors Causing Vacant Homes in Asan City from a Local Perspective

The analysis of the models from a global perspective generally indicated low explanatory power, leading to the conclusion that additional explanatory models were needed based on the characteristics of different regional types. Therefore, a localized analysis of the factors causing vacant homes was conducted, targeting two types of areas identified as having a high concentration of vacant homes: "Old Town type" and "Rural Village". As shown in Table 5, All seven variables utilized in the global model were applied to construct and analyze the Geographically Weighted Regression (GWR) model.

**Table 5.** Analysis results of GWR.

| Explanatory Variable | NBRM Coefficient | GWR Coefficient | | |
|---|---|---|---|---|
| | | AVG Asan | Old Town | Rural Village |
| Vacant Stores | 0.029 ** | −0.0111 | 0.0093 | −0.0692 |
| Undersized Plot Ratio | 5.393 *** | 0.5665 | 0.9532 | −0.3070 |
| Average Number of Floors | 0.176 *** | 0.1006 | 0.0036 | 0.3967 |
| Total Pop Change Rate | 0.005 * | −0.0043 | −0.0032 | −0.0083 |
| Distance to New Development | −0.00007 *** | −0.0001 | −0.0001 | −0.0001 |
| Num. of Construction Workers | 0.004 | −0.0065 | −0.0058 | −0.0106 |
| Land Use Mix | −0.008 | 0.0918 | 0.1118 | 0.0598 |

*: *p*-value < 0.1, **: *p*-value < 0.05, ***: *p*-value < 0.01.

The regression coefficients of the seven explanatory variables acted as different influencing factors in different regions, confirming that the factors affecting the occurrence of vacant homes exhibited spatial heterogeneity at the regional level. The details are as shown

in Figure 6. Comparing the local regression coefficients for factors influencing vacant home occurrences by type, the Old Town type showed the highest coefficients for the undersized plot ratio and land use mix, while the Rural Village type showed the highest for average number of floors and land use mix. Examining the degree of influence of the explanatory variables predicted to be the main factors in the occurrence of vacant homes in the Old Town type, it was found that a 1% increase in population decreased the number of vacant homes by 0.003, a 1 km increase in distance from new development projects decreased the number of vacant homes by 0.1, and an increase of one construction worker decreased the number of vacant homes by 0.006.

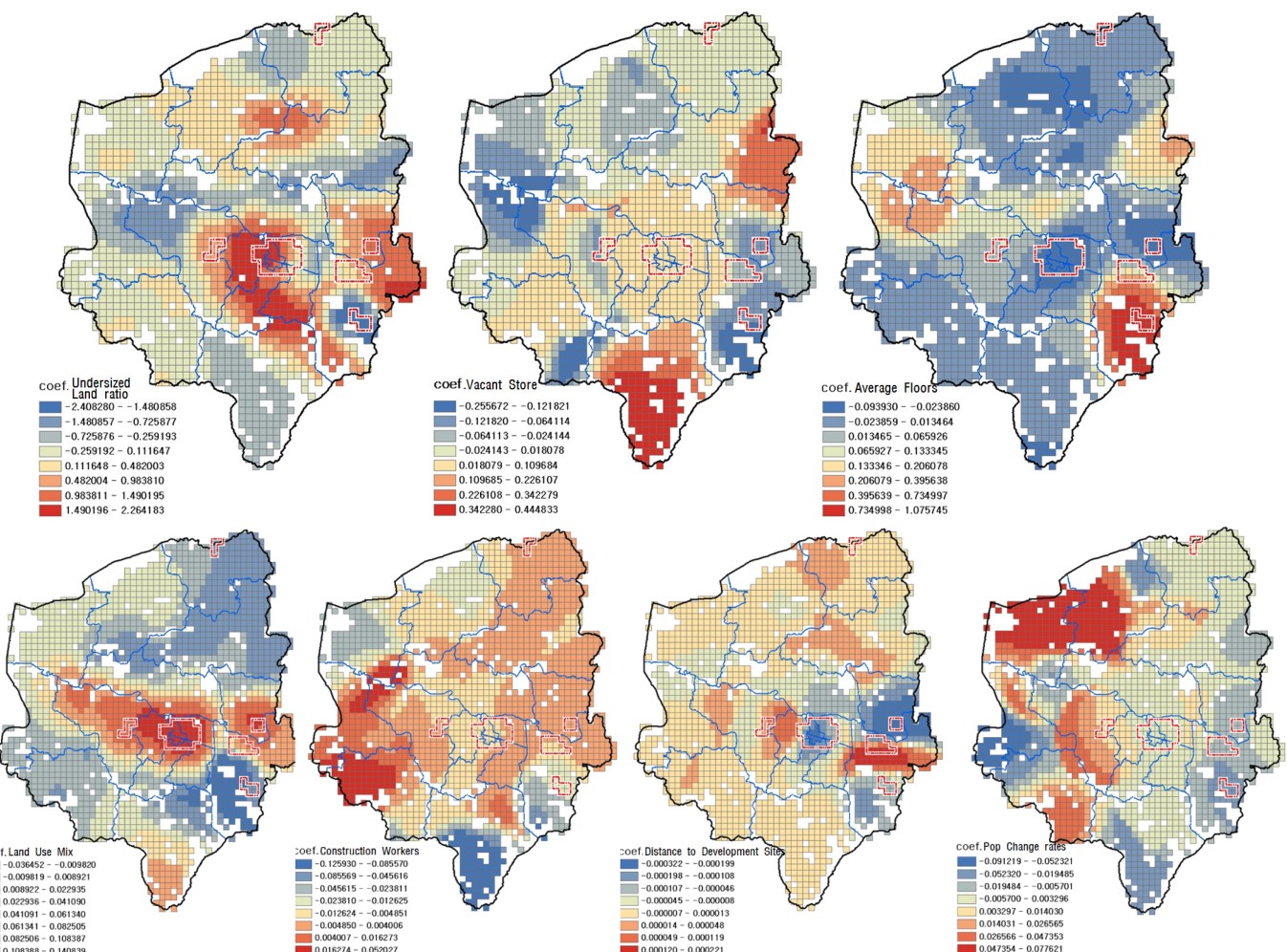

**Figure 6.** GWR Coefficients by Factors.

*5.4. Results and Discussion*

This study identifies significant factors influencing the occurrence of vacant homes in Asan City through the use of spatial and regression analysis. From a localized perspective, the increase in the total population growth rate significantly mitigates the prevalence of vacant homes. This suggests that regions experiencing population growth have better economic vitality, which discourages the abandonment of properties. Conversely, areas witnessing population decline or stagnation tend to see an uptick in vacant homes. This aligns with findings from other studies which show that vibrant, growing populations are essential for maintaining healthy urban environments. Avoiding new development projects in areas already suffering from high vacancy rates is crucial. Such strategies prevent further exacerbation of home vacancies by not diluting the market with additional unneeded housing stock [39]. This approach aligns with sustainable urban planning principles that prioritize infill development and the utilization of existing urban spaces before expanding

outward. Moreover, an increase in the economically active population within a region's main industries appears to directly decrease the occurrence of vacant homes [40]. This finding highlights the importance of economic stability and job availability in preventing urban decline, suggesting that policies aimed at boosting local economies could serve as effective measures against increasing vacancies. The study also found that the number of vacant stores and the Undersized Plot Ratio are influenced by different factors depending on the regional characteristics, underscoring the necessity of applying geographically weighted regression analysis. This method allows for the nuanced understanding of how local conditions influence vacancy rates, providing a more accurate tool for urban planners and policymakers aiming to tackle this issue effectively.

These findings contribute to the field of urban sustainability by demonstrating that the management of vacant homes requires a multifaceted approach. Strategies should not only focus on the economic revitalization of cities but also consider social and environmental factors that contribute to a holistic urban development strategy. Implementing policies that support population growth, economic stability, and the optimal use of land are critical in ensuring the sustainability of medium-sized cities like Asan. Moreover, this research supports the idea that urban planning should be responsive to the specific needs and conditions of different regions within a city. Generic, universal approaches are less effective than strategies tailored to the unique characteristics of each area.

## 6. Conclusions

Korea is facing the reality of a keyword known as "regional extinction". The phenomenon of shrinking cities, predominantly affecting regional urban areas, coupled with the rise in vacant homes within original downtown sectors due to insufficient post-regeneration efforts, and growing concern over escalating numbers of vacant properties in existing cities triggered by central government-driven housing expansion initiatives, represent critical challenges that demand our attention. To prepare countermeasures for the vacant home problem caused by urban decline, an empirical analysis study was conducted targeting Asan city, one of the medium-sized cities in the provinces. The study thoroughly reviewed the relationship between urban decline and the occurrence of vacant homes, the characteristics of development and decline in large and medium cities, and the factors and impacts of vacant home occurrence in large and medium cities, proceeding with a status analysis, the selection of explanatory variables, and an analysis of causative factors.

In Asan city, 266 vacant homes were identified, and through dot distribution patterns and a regional characteristic analysis, the spatial types of areas with a high concentration of vacant homes were defined as "Old Town type" and "Rural Village type". Considering the population, industry, and location characteristics of Asan city, and factors causing vacant homes, as presented in previous studies, 36 explanatory variables were derived and categorized into internal and external factors. The analysis of factors causing vacant homes from a global perspective in Asan city revealed that vacant stores, the average number of floors, total population change rate, distance to new development projects, and land use mix are factors triggering vacant homes. Our Geographically Weighted Regression (GWR) analysis identified that, from a localized perspective, the undersized plot ratio and land use mix significantly influence vacant home occurrences in 'Old Town' areas, whereas in 'Rural Village' areas, the predominant factors are the average number of floors and land use mix. The favorable population and industrial conditions unique to Asan city, differentiating it from other medium cities, also act as negative influencing factors on the occurrence of vacant homes.

Our analysis of the population growth patterns in the downtown and suburban areas of Asan city indicated that Asan city is in the suburbanization stage according to Berg's (1982) urban development stages and should be considered a growing city. However, despite this, the occurrence of vacant homes due to the decline of some areas, such as the original downtown, is problematic, and if it reaches the stage of deurbanization, where both the downtown and outer populations decrease, the idle real estate problem could be

very difficult to address. Despite acting as a factor reducing vacant homes in some areas due to the solid population and industrial base in Asan city, physical environments (such as undersized plots and land use mix) have a significant impact on the occurrence of vacant homes. In the paradigm of shrinking cities where low birth rates, aging, and low growth are becoming entrenched, policies should be pursued with the goal of reducing vacant stores, promoting natural and social increases in population, and reusing existing facilities rather than pursuing new development projects to prevent the occurrence of idle real estate, including vacant homes, within the city. At the same time, quick revitalization through the designation of maintenance areas is necessary for regions like "Old Town" and "Rural Village", where vacant homes are locally concentrated. In a place like Asan city, it is crucial to prevent the regional decline caused by a few vacant homes through active vacant home maintenance, e.g., investment in infrastructure such as roads and parks.

Previous studies on vacant homes have analyzed the factors of vacancy on a global scale, targeting different areas such as large cities, medium-sized cities, and rural areas. However, this study analyzed both global and localized factors within a single city. Through this approach, it has provided an opportunity to simultaneously consider vacant home policies at the city level and for specific areas within the city. Through this research, we hope to promote policies that proactively address the issue of vacant homes at the city level, thereby enhancing urban sustainability.

**Author Contributions:** Conceptualization, J.-h.C.; Data Curation, J.-h.C.; Formal analysis, J.-h.C. and S.-S.H.; Investigation and Methodology, J.-h.C. and M.-j.W.; Project administration, J.-h.C.; Resources, Software, and Supervision, S.-S.H. and M.-j.W.; Validation and Visualization: J.-h.C.; Writing, J.-h.C. All authors have read and agreed to the published version of the manuscript.

**Funding:** This research received no external funding.

**Institutional Review Board Statement:** Not applicable.

**Informed Consent Statement:** Not applicable.

**Data Availability Statement:** Data is contained within the article, and Available from the Korea National Statistical Portal.

**Conflicts of Interest:** The authors declare no conflict of interest.

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
