# Peer review of "Analyzing the Factors of Vacant Home Occurrence for Urban Sustainability: A Case Study of Medium-Sized Cities Focusing on Asan City, Chungcheongnam-do"

_sustainability, doi:10.3390/su16093742_

Round 1
Reviewer 1 Report
Comments and Suggestions for Authors
This study is to spatially analyze the characteristics of vacant homes and identify the factors leading to vacant homes. The idea is good and attractive. However, I believe further consideration is warranted regarding the alignment between the title and the content. And there are still problems that need to address before it meets publishing standards.

Comments on the Quality of English Languagesee above
Author Response
Thank you for your valuable feedback.
Please see the attachment.

Reviewer 2 Report
Comments and Suggestions for Authors
The subject is interesting and the paper is well-written. The methodology is correct. Just one minor remark - please consider proofreading of the paper by the Authors to eliminate minor laguage issues.

Author Response

(The authors gave the same response as above.)

Reviewer 3 Report
Comments and Suggestions for Authors
Vacant homes are certainly a problematic phenomenon. Therefore, the undertaken research is important and timely. The article is chaotic and needs to be rearranged. Below are some comments that will help improve the article:
1. The purpose of the research should be clearly defined (now it is different in the abstract and the introduction).
2. A more extensive literature review should be conducted.
3. There is a lack of scientific discussion related to the conducted research.
4. In line 396, the authors suggest that there are many factors determining the incidence of vacant houses and that other studies confirm it. Similarly, the first paragraph of section 3.2 suggests many previous studies, and there is a reference to one item. Please refer to other researchers.
5. The part of the introduction from line 45 is more will be in the methodology.
Author Response

(The authors gave the same response as above.)

Round 2
Reviewer 1 Report
Comments and Suggestions for Authors
no suggestions
Comments on the Quality of English Languageseems like to need further improvement
Reviewer 3 Report
Comments and Suggestions for Authors
Thank you for your comprehensive answers. In my opinion, the scientific discussion should be in paragraph 5. The authors should complete this chapter.
With the additions made, the article has gained in quality.
